# GRAPE: Let GRPO Supervise Query Rewriting by Ranking for Retrieval

**Zhaohua Zhang** [1 2]   **Jianhuan Zhuo** [2]   **Muxi Chen** [2]   **Chenchen Zhao** [2]   **Wenyu Jiang** [2]   **Mingyang Chen** [2]
**Yu Tang** [2]   **Qiuyong Xiao** [2]   **Tianwen Jiang** [2]   **Jihong Zhang** [2]   **Zhixun Su** [1]

## Abstract

The CLIP model has established itself as a cornerstone of large-scale retrieval systems. However, its performance often degrades under distributional shifts such as multilingual, long-form, or multimodal queries. To avoid the prohibitive costs associated with retriever retraining or corpus re-embedding, we propose GRAPE (**G**rouped **R**anking-**A**ware **P**olicy Optimization **E**nhancement), a plug-and-play approach that leverages LLM-based query rewriting to bridge these gaps. Unlike existing methods that lack explicit supervision, GRAPE integrates ranking signals into the rewriting LLM via Grouped Relative Policy Optimization (GRPO), ensuring rewritten queries are better aligned with the frozen retriever's latent distribution. Crucially, we identify a *score inflation* phenomenon in naive similarity-based finetuning—where irrelevant candidates receive indiscriminately high scores—and mitigate it with a novel corpus-relative ranking-based reward. Extensive experiments across multilingual (Flickr30k-CN, CVLUE, XM3600), long-form (Wikipedia), and multimodal (CIRR) benchmarks demonstrate that GRAPE consistently improves performance, achieving an average gain of 4.9% in Recall@10 without any modification to the underlying retriever. The code is available at https://github.com/mogulzhang/GRAPE.

## 1. Introduction

Contrastive Language–Image Pretraining (CLIP) (Radford et al., 2021) has emerged as a cornerstone of modern vision-language systems by mapping multimodal data into a unified, shared semantic space. Due to its remarkable efficiency and versatility, vector indexes on a scale of tens-of-billions (Chuang et al., 2025; Wang et al., 2025a) have been deployed across industrial sectors. In these ecosystems, CLIP serves as the pivotal encoder for a wide array of downstream tasks, ranging from cross-modal retrieval (Stevens et al., 2024) to zero-shot classification (Qu et al., 2025b; Martin et al., 2024; Wu et al., 2025) and unsupervised clustering (Qu et al., 2025a).

Nevertheless, as the scope of retrieval-oriented applications continues to broaden, the inherent limitations of these frozen encoders become apparent. Specifically, system performance frequently degrades when the target input distributions diverge significantly from CLIP's pre-training corpus—a challenge commonly observed in scenarios involving multilingual queries, long-form descriptions, or complex multimodal shifts.

A conventional strategy to mitigate these shifts involves expanding the pre-training corpus to encompass downstream distributions (Fan et al., 2023; Huang et al., 2026; Yuksekgonul et al., 2022), followed by domain-specific finetuning to yield performance gains. However, a significant drawback of retraining is that it inevitably alters the learned embedding space, necessitating the costly re-embedding of all existing corpus data and the redeployment of downstream applications. Given the sheer scale of industrial vector databases, such re-indexing is often prohibitively expensive and operationally disruptive. Consequently, enhancing existing CLIP models while preserving the integrity of the established retrieval infrastructure has become a critical and practical challenge.

A promising alternative is to leave the retrieval pipeline untouched and instead bridge distribution gaps from the query side by leveraging Large Language Models (LLMs) for query rewriting. These methods typically employ prompt engineering to guide LLMs in transforming downstream queries into forms that better align with CLIP's original training distribution. Nevertheless, such zero-shot rewriting remains suboptimal, as LLMs lack explicit knowledge of the retriever's latent space and therefore cannot consistently generate high-quality rewrites. While subsequent

---

[1]Dalian University of Technology, Dalian, China [2]Tencent Hunyuan AI Data, Shenzhen, China. Correspondence to: Zhixun Su <zxsu@dlut.edu.cn>, Jihong Zhang <jihongzhang@tencent.com>, Tianwen Jiang <tianwjiang@tencent.com>, Jianhuan Zhuo <tomzhuo@tencent.com>.

*Proceedings of the 43rd International Conference on Machine Learning*, Seoul, South Korea. PMLR 306, 2026. Copyright 2026 by the author(s).

works have attempted to address this—for example, via fine-grained decomposition (Jiang et al., 2022) or interactive user feedback (Lee et al., 2024)—they only partially resolve distribution mismatches and often rely on inefficient, heuristic-based rules. Therefore, we argue that it is essential to design an approach that can distill feedback signals directly from the retriever to supervise the query-rewriting process in a more reliable and data-driven direction.

Drawing inspiration from Group Relative Policy Optimization (GRPO) (Shao et al., 2024), we address these challenges with GRAPE (**G**rouped **R**anking-**A**ware **P**olicy Optimization **E**nhancement)—a seamless, efficient approach that integrates retrieval-derived ranking signals into LLM-based query rewriting. GRAPE leverages the grouped optimization paradigm to bridge diverse distributional gaps—spanning language, length, and modality shifts—by transforming raw queries into representations that better align with the retriever's original training distribution.Crucially, our empirical analysis reveals that directly finetuning LLMs via similarity scores induces a score inflation phenomenon, where the model generates rewrites that receive indiscriminately elevated scores regardless of their actual retrieval relevance. To mitigate this, we propose a corpus-relative ranking-based reward. This mechanism explicitly aligns the optimization objective with ranking metrics, effectively suppressing spurious score inflation while prioritizing discriminative power. Extensive evaluations across multilingual (Flickr30k-CN, CVLUE, XM3600), long-form (Wikipedia), and multimodal (CIRR) benchmarks demonstrate that GRAPE consistently enhances performance, yielding an average improvement of 4.9% in Recall@10.Our primary contributions are summarized as follows:

- We propose GRAPE, a plug-and-play retrieval enhancement approach that keeps the retriever frozen. By optimizing query rewriting through retrieval-guided feedback, GRAPE obviates the need for costly model retraining or corpus re-embedding.

- We identify the score inflation pitfall in similarity-based supervision and introduce a ranking-based reward function. This approach aligns optimization with actual ranking objectives, ensuring the generation of high-fidelity, discriminative rewrites.

- We demonstrate that GRAPE achieves significant and consistent gains across five representative benchmarks, addressing multilingual, length, and multimodal shifts with an average Recall@10 improvement of 4.9%.

## 2. Related Works

### 2.1. Training-based Retrieval

Training-based methods address the issue of input distributions diverging substantially from CLIP's training corpus by expanding the training data and subsequently finetuning the model. These methods generally fall into two categories: data-centric approaches, which focus on scaling or refining the training corpus, and model-centric approaches, which develop more sophisticated architectures to improve representation learning and cross-modal alignment.

**Data-centric approaches.** These methods focus on expanding or refining the training corpus. Large-scale efforts such as MetaCLIP-2 (Chuang et al., 2025) and 100B-scale pre-training (Wang et al., 2025a) achieve notable gains in cross-lingual retrieval but demand massive resources. Other studies emphasize data quality, such as rewriting captions for richer supervision (Fan et al., 2023) or introducing composition-aware hard negatives to mitigate bag-of-words bias (Yuksekgonul et al., 2022). Although effective, these approaches are costly to construct and preprocess, and their improvements often remain confined to limited downstream tasks.

**Model-centric approaches.** These methods focus on optimizing internal representations or addressing structural constraints. Representative examples include BLIP-2 (Li et al., 2023), which introduces a trainable Q-Former to bridge frozen encoders with LLMs; finetuning strategies for pairwise reasoning (Sam et al., 2024) or compositional retrieval (Baldrati et al., 2023); and parameter-efficient tuning techniques such as adapters and LoRA (Wang et al., 2023). Other lines of work explore prompt learning (Zheng et al., 2025), long-text handling (Long-CLIP (Zhang et al., 2024), LoTLIP (Wu et al., 2024), FineLIP (Asokan et al., 2025)), multilingual adaptation (mCLIP (Chen et al., 2023), AltCLIP (Chen et al., 2022)), and multimodal fusion (CIR (Baldrati et al., 2023)). While these approaches achieve strong task-specific improvements, they inevitably incur retraining overhead and require re-generating large-scale pre-computed embedding databases.

### 2.2. Training-free Retrieval

Training-free methods aim to enhance CLIP's retrieval capability without modifying the encoder or regenerating large-scale embeddings, instead relying on external strategies that can be seamlessly integrated into existing systems. A representative direction focuses on query rewriting and augmentation. ComCLIP (Jiang et al., 2022) adopts a framework that decomposes both images and texts into semantic sub-parts and then aligns these components to achieve fine-grained alignment, but it is limited to queries and images whose semantics can be effectively decomposed. More recently,

PlugIR (Lee et al., 2024) employs LLMs to iteratively refine user queries through dialogue and candidate context. However, owing to the lack of supervision, this approach relies on substantial hand-crafted constraints, which in turn limits its practicality in real-world deployments.

To address this, we propose GRAPE, a plug-and-play framework that leverages ranking-aware LLM rewriting for robust distribution adaptation without retriever retraining.

## 3. Method

### 3.1. Problem Formulation.

In CLIP-based retrieval tasks, given a query $q \sim \mathcal{D}_{\text{query}}$, the system embeds the query and each candidate item $i \in \mathcal{I} = \{i_1, \ldots, i_N\}$ into a shared semantic space and ranks candidates by similarity scores.

For text-to-image retrieval, let

$$z_q = f_{\text{text}}(q), \qquad z_i = f_{\text{img}}(i) \qquad (1)$$

where $f(\cdot)$ denotes the CLIP encoder, $z_q$ and $z_i$ are $\ell_2$-normalized embeddings. The top-ranked item is selected by

$$i^\star = \arg\max_{i \in \mathcal{I}} \ s(z_q, z_i) \qquad (2)$$

where $s(\cdot, \cdot)$ denotes the similarity function (e.g., cosine similarity). While CLIP achieves strong performance when queries follow its training distribution $\mathcal{D}_{\text{train}}$, it struggles under distributional shifts between the query distribution $\mathcal{D}_{\text{query}}$ and $\mathcal{D}_{\text{train}}$ (e.g., query length, multilingual inputs, or modality differences).

To address this issue, one approach is to expand $\mathcal{D}_{\text{train}}$ and finetune the retriever, but this incurs prohibitively high costs due to re-embedding and redeployment. An alternative is to leverage LLM knowledge to transform a query $q$ into a rewritten query $\tilde{q}$ that better matches $\mathcal{D}_{\text{train}}$. However, under the absence of supervision signals, such rewriting cannot fully align $\mathcal{D}_{\text{query}}$ with $\mathcal{D}_{\text{train}}$.

Therefore, our objective is to incorporate supervision signals into the distribution adaptation process, thereby enabling more effective query rewriting and bridge the distributional shifts.

### 3.2. GRAPE: Grouped Ranking-Aware Policy Optimization Enhancement

To address these challenges, we introduce GRAPE, a plug-and-play enhancement approach that incorporates ranking signals through retrieval-guided query rewriting with LLMs. Specifically, GRAPE rewrites a query $q$ into a group of queries $\tilde{q}_1, \ldots, \tilde{q}_k$ using an LLM, and leverages the relative ranking of the target image obtained from this group

to update subsequent rewrites. The overall architecture is illustrated in Figure 1.

GRAPE operates in two phases: (i) Query Rewriting and Retrieval, where each query is rewritten into a group of queries and the frozen retriever yields the rankings of the target image; and (ii) Relative Reward Computation and Policy Optimization, where ranking results are used to compute relative rewards within the group and to optimize the rewriting LLM.

**Query Rewriting and Retrieval.** Given a raw query $q$, an LLM policy $\pi_\theta$ samples $K$ constrained rewrites $\{\tilde{q}_k\}_{k=1}^{K}$. After format validation, each rewrite is embedded by the *frozen* CLIP text encoder, $z_{\tilde{q}_k} = f_{\text{text}}(\tilde{q}_k)$; we then compute $s(z_{\tilde{q}_k}, z_i)_{i \in \mathcal{I}}$ against the precomputed candidate embeddings, rank all items accordingly, and obtain the target image rankings $\{r_k\}_{k=1}^{K}$.

**Relative Reward Computation and Optimization.** For every rewrite, GRAPE constructs two complementary reward. *Format reward:* Reasoning must be inside `<think>...</think>` and the final rewrite must be inside `<answer>...</answer>`. Non-conforming outputs receive $R^{\text{f}} = -1$ and are skipped (no CLIP retrieval); conforming outputs receive $R^{\text{f}} = 1$ and proceed to retrieval. *Ranking reward:* $R^{\text{r}}$ measures retrieval effectiveness using the rankings $\{r_k\}_{k=1}^{K}$. It is assigned after retrieval and increases monotonically with ranking quality (details in Section 3.3)

We integrate the two rewards into a unified feedback signal:

$$R_k = R_k^{\text{f}} + R_k^{\text{r}},$$

and compute group-wise statistics over the $K$ rewrites of the same query $q$:

$$\mu_q = \frac{1}{K} \sum_{k=1}^{K} R_k, \qquad \sigma_q^2 = \frac{1}{K} \sum_{k=1}^{K} \left( R_k - \mu_q \right)^2.$$

The relative advantage is

$$\tilde{A}_k = \frac{(R_k^{\text{f}} + R_k^{\text{r}}) - \mu_q}{\sqrt{\sigma_q^2}},$$

which emphasizes within-group improvements while stabilizing scale. The policy is updated with an advantage-weighted objective regularized toward a reference model $\pi_{\text{ref}}$:

$$\begin{aligned}
\mathcal{J}(\theta) =& \mathbb{E}_{q \sim \mathcal{D}} \left[ \frac{1}{K} \sum_{k=1}^{K} \tilde{A}_k \log \pi_\theta(r_k \mid q, \mathcal{C}) \right] \\
& - \lambda \, \mathbb{E}_{q \sim \mathcal{D}} \left[ \text{KL}\big( \pi_\theta(\cdot \mid q, \mathcal{C}) \, \| \, \pi_{\text{ref}}(\cdot \mid q, \mathcal{C}) \big) \right],
\end{aligned} \qquad (3)$$

where only $\pi_\theta$ is updated; the CLIP encoders and corpus embeddings remain frozen.

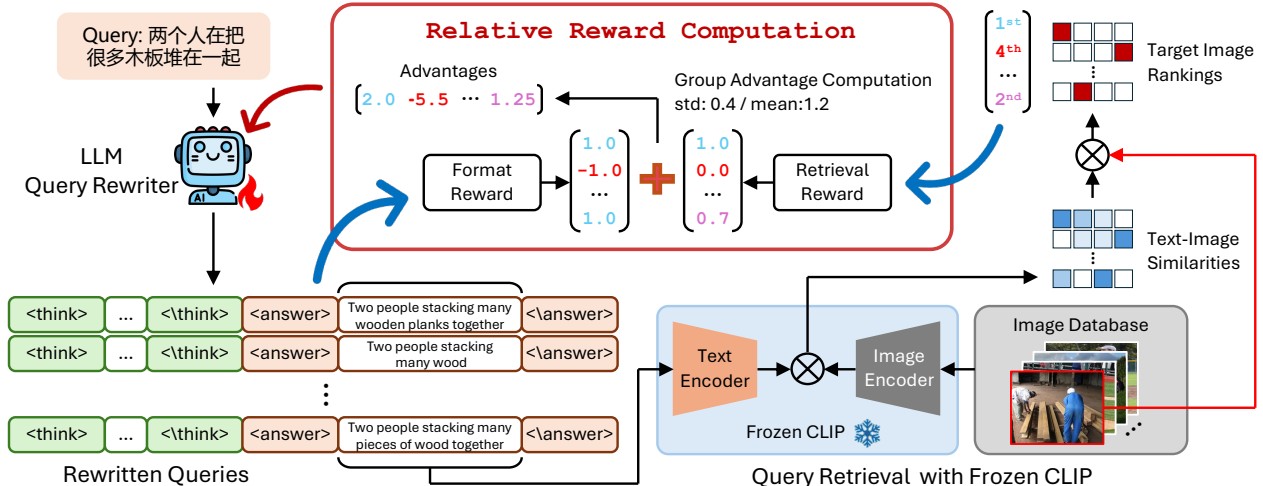

*Figure 1.* **Overview of GRAPE.** The approach operates in two phases: (i) *Query Rewriting and Retrieval*, where an LLM generates $K$ rewrites for a query, each encoded by a frozen CLIP retriever to produce target ranks; and (ii) *Relative Reward Computation and Optimization*, where a format reward and a rank reward are combined into a unified feedback signal, which is normalized within group to compute relative advantages and then used to update the LLM.

### 3.3. Why Ranking Beats Score: Relative Ranking Reveals True Relevance

**Ranking-Based Reward Function.** The rank of the target image serves as the most direct proxy for retrieval quality, providing an intrinsic signal for evaluating the efficacy of LLM-generated rewrites. Unlike similarity-based metrics, ranking reflects the relative discriminative power of a query within the candidate corpus. We therefore propose a ranking-based reward function to supervise the rewriting policy:

$$R_k^{\mathrm{r}} \;=\; 1 \,-\, \frac{2\,(r_k - 1)}{N - 1},\qquad(4)$$

where $r_k \in \{1,\dots,N\}$ denotes the rank of the target image among $N$ candidates.

As illustrated in Figure 1, for a set of candidate rewrites $\{\tilde{q}_k\}_{k=1}^{K}$ sampled from the policy, those that successfully capture retrieval-essential semantic details yield superior rankings and, consequently, higher rewards. These informative signals reinforce the model to refine its policy toward generating rewrites that prioritize retrieval-centric alignment. Moreover, the linear mapping in Eq. 4 ensures that the reward is sensitive to every incremental change in ranking across the candidate space. A detailed derivation and analysis of the reward are provided in Appendix A.1.

**Comparison with Similarity-Based Rewards.** While similarity scores offer a straightforward metric for query-target proximity, monotonic increases in similarity do not inherently translate to improved retrieval rankings. For instance, the inclusion of non-discriminative tokens—such as

"real-world" or "environment"—can boost absolute similarity scores for the target image while simultaneously increasing similarity with numerous irrelevant candidates. Consequently, under the similarity-based reward

$$R_k^{\mathrm{s}} \;=\; s(z_{\tilde{q}_k}, z_t)\,,\qquad(5)$$

the finetuned LLM often suffers from **score inflation**, where nearly all candidates are assigned indiscriminately high scores regardless of their true relevance. In practice, the model exploits this by injecting high-frequency but low-specificity tokens that increase semantic overlap without enhancing discriminative power. Although the model receives positive reinforcement, these signals fail to steer the optimization toward an effective retrieval-aware trajectory.

**Direct Effect on Optimization.** By computing relative advantages within each group, the primary optimization signal shifts from absolute score gaps to **relative ranking dynamics**. This mechanism ensures that the driving force behind policy updates is the relative improvement in ranking order rather than absolute similarity. Consequently, GRAPE guides the LLM to generate rewrites that promote genuine retrieval performance, effectively mitigating the degenerate behavior associated with similarity-driven score inflation.

## 4. Experiments

In this section, we evaluate the effectiveness of our approach in addressing three major distributional differences commonly induced by queries in retrieval tasks: (1) **multilingual differences**—queries in languages that are under-

represented in CLIP's pretraining corpus; (2) **length differences**—query styles (e.g., long-form or complex phrasing) that deviate from the training distribution; and (3) **modality differences**—multimodal or compositional queries requiring reasoning beyond text-only inputs. Furthermore, to demonstrate the *data efficiency* of our method, we conduct experiments under varying proportions of training data. We also analyze the impact of similarity-based score inflation, verifying that our proposed ranking-based reward effectively suppresses this artifact.

### 4.1. Experimental Setup

**Datasets.** To evaluate the effectiveness of our method, we conduct experiments on five representative benchmarks, each designed to target one or more of the core challenges discussed above: multilingual differences, length differences, and modality differences.

- **Flickr30k-CN** (Lan et al., 2017): A Chinese extension of Flickr30k, designed for cross-lingual image–text retrieval.

- **CVLUE** (Wang et al., 2025b): A large-scale Chinese multilingual benchmark . It primarily addresses multilingual difference and Chinese culture understanding.

- **XM3600** (Thapliyal et al., 2022): A multilingual benchmark covering 36 languages, aimed at testing cross-lingual generalization and robustness.

- **Wikipedia** (Rasiwasia et al., 2010): An English text-to-image retrieval dataset featuring long and complex queries, making it suitable for evaluating length differences.

- **CIRR** (Liu et al., 2021): A fine-grained, compositional retrieval benchmark where both image and text serve as queries. It emphasizes modality discrepancies and requires nuanced semantic understanding. We follow an 80%/20% train/validation split for our experiments.

**Comparison Methods.** We validate the scalability of our method across three CLIP variants: ViT-B/32, ViT-B/16, and ViT-L/14. For each backbone and dataset, we horizontally compare the performance of:

- **CLIP (baseline)**: frozen pretrained CLIP without query rewriting.

- **CLIP+LLM**: frozen CLIP with queries rewritten by a frozen LLM without retrieval feedback.

- **CLIP+GRPO-LLM**: frozen CLIP with queries rewritten by an LLM finetuned with GRPO, where training is conducted on the training set of the corresponding dataset using retrieval-based rewards.

**Evaluation Metrics.** To achieve a more accurate evaluation, we calculate two key metrics respectively: the proportions of cases where the target image is included in the top-1 and top-10 retrieved results (i.e., R@1 and R@10).

**Implementation and Training Setup.** We employ `Qwen2.5-3B-Instruct` as the primary rewriter, utilizing `Qwen2-VL-7B` specifically for the multimodal tasks in CIRR. Three task-specific prompt templates are applied consistently across both training and inference to ensure evaluation integrity. GRAPE is optimized over 1,500–2,500 steps—depending on dataset scale—via AdamW with a $5 \times 10^{-7}$ learning rate, a cosine decay schedule, and a 0.03 warmup ratio. Experiments are conducted with a batch size of 8 using gradient accumulation. We evaluate generalizability across three frozen CLIP retrievers: ViT-B/32, ViT-B/16, and ViT-L/14. Detailed configurations are provided in Appendix A.1.

### 4.2. Main Results

As shown in Table 1, the distribution gap between training data and queries limits CLIP's ability to handle cross-lingual and multimodal inputs, and even for long-text queries the recall rate remains relatively low. While query rewriting with LLMs can partially mitigate this gap, the absence of optimization guidance often leads to suboptimal rewrites. In contrast, our proposed GRAPE framework effectively aligns query distributions with the retriever, yielding consistent improvements across all settings. On average, GRAPE achieves gains of more than 4.5% in R@1 and 4.9% in R@10 over CLIP+LLM, and these improvements hold across different CLIP variants, with stronger models achieving higher absolute recall under GRAPE enhancement.

### 4.3. Critical Analysis

Why can GRAPE address the three major distributional differences commonly induced by queries in retrieval tasks? To answer this, we conduct analytical experiments on how GRAPE overcomes the challenges that arise when adapting query distributions: multilingual differences, by performing cross-lingual translation and semantic enrichment; length differences, by conducting long-text distillation and expansion; and modality differences, by enabling cross-modal understanding and generating retrieval-friendly expressions.

**Cross-Lingual Query Translation and Semantic Enrichment.** This challenge arises not only from the difficulty of accurate translation, but also because direct translations often overlook implicit semantics required for retrieval, resulting in literal yet incomplete queries. As illustrated in Fig. 2(a), the original query implicitly refers to a "baseball player," yet a vanilla LLM produces only a literal translation that misses this hidden entity. Similarly, in Fig. 2(b),

*Table 1.* R@1 and R@10 for different CLIP model scales and methods. "+LLM" indicates frozen LLM rewriting, while "+GRAPE" denotes GRPO-finetuned rewriting (ours). GRAPE consistently achieves notable improvements over both vanilla CLIP and CLIP+LLM across all benchmarks and model sizes, delivering significant gains in R@1 and R@10. A dash ("–") indicates that the corresponding model is not capable of handling the task directly.

| Model | Flickr30k-CN T2I | | CVLUE T2I | | XM3600 T2I | | Wikipedia T2I | | CIRR TI2I | | Average | |
|---|---|---|---|---|---|---|---|---|---|---|---|---|
| | R@1 | R@10 | R@1 | R@10 | R@1 | R@10 | R@1 | R@10 | R@1 | R@10 | R@1 | R@10 |
| ViT-B/32 | — | — | — | — | 11.4 | 25.5 | 28.4 | 65.8 | — | — | 19.9 | 45.6 |
| +LLM | 49.8 | 84.4 | 10.6 | 38.5 | 49.5 | 77.8 | 30.4 | 67.1 | 29.5 | 70.5 | 34.0 | 67.7 |
| **+GRAPE(↑)** | **53.3** | **87.3** | **13.1** | **44.4** | **56.0** | **83.1** | **36.1** | **78.4** | **34.3** | **77.8** | **38.6** | **74.2** |
| ViT-B/16 | — | — | — | — | 12.3 | 26.8 | 33.5 | 71.9 | — | — | 22.9 | 49.4 |
| +LLM | 52.1 | 86.0 | 12.1 | 42.0 | 52.9 | 78.9 | 34.3 | 69.3 | 30.7 | 73.3 | 36.4 | 69.9 |
| **+GRAPE(↑)** | **58.0** | **90.0** | **14.6** | **48.7** | **58.4** | **83.1** | **43.7** | **82.8** | **35.7** | **77.8** | **42.1** | **76.5** |
| ViT-L/14 | — | — | — | — | 14.2 | 29.0 | 40.4 | 80.2 | — | — | 27.3 | 54.6 |
| +LLM | 57.6 | 89.5 | 13.7 | 43.6 | 51.7 | 78.1 | 42.0 | 77.6 | 29.7 | 70.1 | 38.9 | 71.9 |
| **+GRAPE(↑)** | **62.6** | **92.6** | **15.3** | **48.9** | **58.1** | **82.6** | **47.8** | **83.5** | **33.2** | **76.3** | **43.4** | **76.8** |

"wooden planks" is simplified to "wood," discarding crucial descriptive details. In contrast, GRAPE generates a retrieval-oriented rewrite that explicitly recover latent entities and enrich semantic details (e.g., "baseball player throwing a ball," "wooden planks"), thereby strengthening cross-lingual rewrite.

**Long-Text Expression Distillation and Expansion.** This challenge arises because long-text queries often contain excessive redundancy that buries the salient concepts, making it difficult for CLIP to capture the true retrieval intent. At the same time, such inputs may also lack explicit emphasis on key entities, leaving gaps in semantic coverage. As shown in Fig. 2(c), a 674-word Wikipedia passage about the film *Mom and Dad* leads a vanilla LLM to generate a verbose 79-word rewrite that even hallucinates additional screening details. In contrast, GRAPE performs both distillation, by removing redundant descriptions, and enrichment, by expanding essential concepts (e.g., *Mom and Dad*, *prints*, *ads*), thus producing concise yet sufficiently informative queries. This synergy between distillation and enrichment enhances retrieval performance on long-text benchmarks.

**Multimodal Understanding and Efficient Expression.** This challenge arises because multimodal queries—which CLIP alone struggles to handle—require the integration of both image and text information. Combining visual and textual inputs demands accurate cross-modal understanding and the generation of efficient representations. However, vanilla VLMs often fail to capture the key cross-modal signals and thus struggle to produce retrieval-friendly expressions. As shown in Fig. 3(a), a vanilla VLM cannot correctly resolve that the textual phrase "similar car" refers to the "Mini Convertible" in the image. As shown in Fig. 3(b),

though fusion occurs, the generated description of "tree" is overly complicated and deviates from the retrieval intent. In contrast, GRAPE learns to effectively integrate multimodal information into unified, retrieval-friendly queries, capturing complementary semantics and thereby enhancing retrieval performance in multimodal scenarios.

### 4.4. Ablation Study

**Data Efficiency.** GRAPE demonstrates remarkable sample efficiency by shifting the learning objective: instead of requiring the LLM to learn new semantic knowledge from static labels, it trains the model to "decode" the latent preferences of the frozen retriever. Unlike conventional Supervised Fine-Tuning (SFT), which relies on vast human-annotated "ground truth" pairs, GRAPE leverages ranking-derived feedback. We evaluate this efficiency on the Flickr30k-CN and CVLUE benchmarks across varying data scales. As illustrated in Fig. 4, GRAPE consistently outperforms the zero-shot LLM baseline across all data regimes. Notably, on the Flickr30k-CN benchmark (Fig. 4a), training with a mere 10k samples—just 10% of the total dataset—allows GRAPE to recover 94% of the Recall@1 performance achieved with the full 100k-sample model. Similarly, on CVLUE (Fig. 4c), the model captures 88% of full-data performance with only 10k samples. This rapid performance saturation confirms that the LLM can distill a robust rewriting strategy from minimal feedback, making GRAPE highly effective for cold-start scenarios in specialized domains where large-scale human supervision is unavailable.

**Limitations of Similarity-Based Supervision.** A critical finding in our analysis is that supervising LLM finetun-

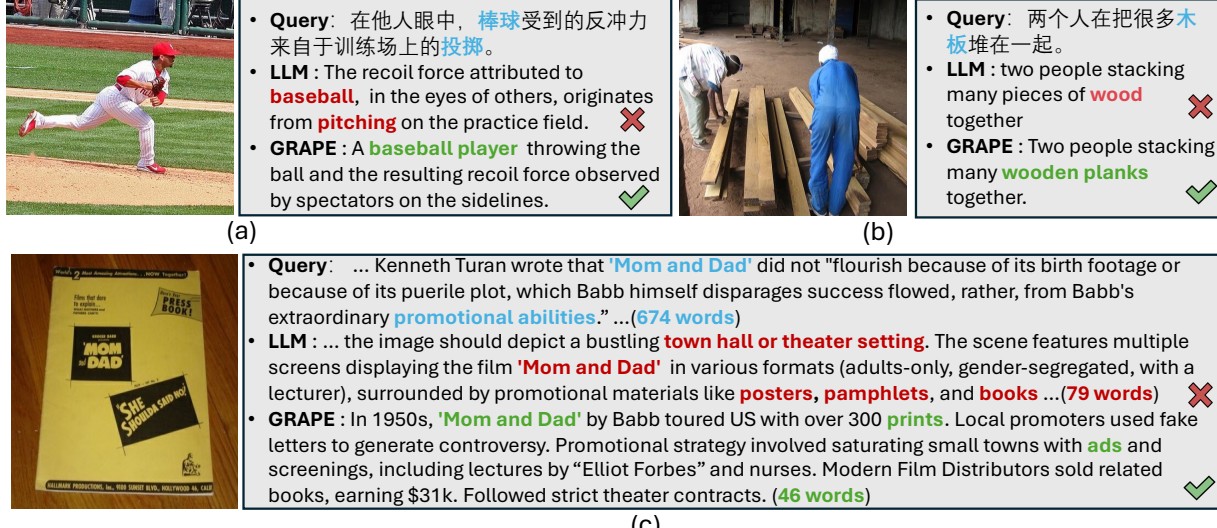

*Figure 2.* Case study of query rewriting, where Figures (a) and (b) illustrate multilingual query cases, and Figure (c) shows a case with a long-form query. The key concepts in the original query are marked blue; literal or incomplete rewrites generated by a vanilla LLM are marked red; retrieval-friendly expressions produced by GRAPE are marked green.

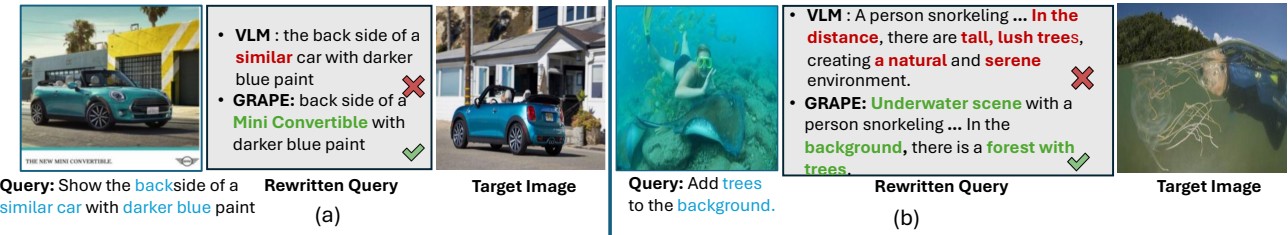

*Figure 3.* Case study of multimodal query rewriting. Since CLIP does not natively support multimodal inputs, image information must first be extracted and then integrated into the text according to the instruction.

ing via direct similarity scores induces a pathological state termed **score inflation**. This phenomenon primarily stems from the inherent flaws of *pointwise supervision*, which evaluates rewritten queries in isolation and fails to account for their relative discriminative utility within the broader retrieval corpus. As evidenced by the empirical analysis on Flickr30k-CN (Fig. 5), we observe a striking decoupling between absolute similarity and actual retrieval accuracy. While the average similarity between rewrites and target images increases monotonically during training, the Recall@1 metrics degrade sharply. Qualitative analysis (see Fig. 5) reveals that the LLM policy frequently adopts an **optimization shortcut** (or reward hacking): it generates high-frequency, non-discriminative tokens—such as real-world environment" or atmosphere and mood"—to boost absolute similarity scores. However, this semantic dilution increases the query's proximity to nearly all candidates in the latent space, thereby diminishing its specificity to the target. Consequently, the rewritten queries become "safe" but uninformative, leading to a collapse in ranking performance.

This failure mode highlights the necessity of our proposed ranking-aware optimization, which forces the LLM to prioritize tokens that enhance visual separability rather than just absolute proximity.

## 5. Discussion

**LLM Knowledge Bottleneck.** The efficacy of GRAPE is fundamentally tied to the LLM's capacity to transform raw queries into forms aligned with the retriever's training distribution. Consequently, performance is intrinsically capped by the LLM's internal knowledge; when downstream tasks exceed this epistemic coverage, the model may struggle to generate semantic-rich or meaningful rewrites. This is particularly evident in multilingual retrieval, where diminished proficiency in low-resource languages constrains the potential gains afforded by GRAPE (see Appendix A.7).

**Retriever-Dependent Ceiling.** GRAPE utilizes ranking signals from a frozen retriever as optimization feedback.

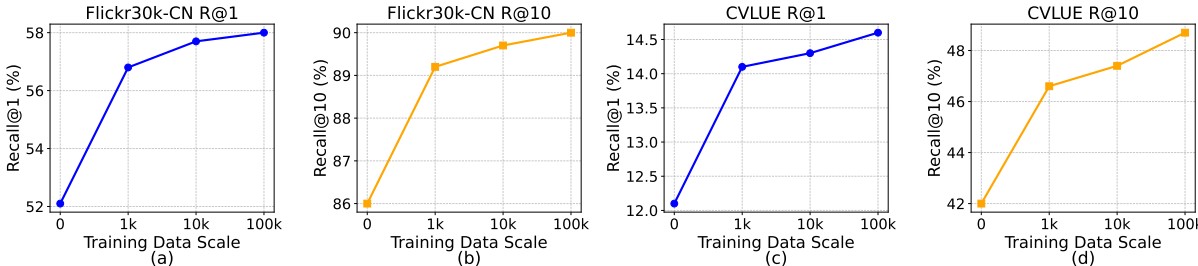

*Figure 4.* **R@1 and R@10 performance across different training data scales.** Results on Flickr30k-CN and CVLUE benchmarks demonstrate GRAPE's data efficiency, achieving competitive performance even with limited samples and consistently surpassing untrained LLM baselines across various distribution shifts.

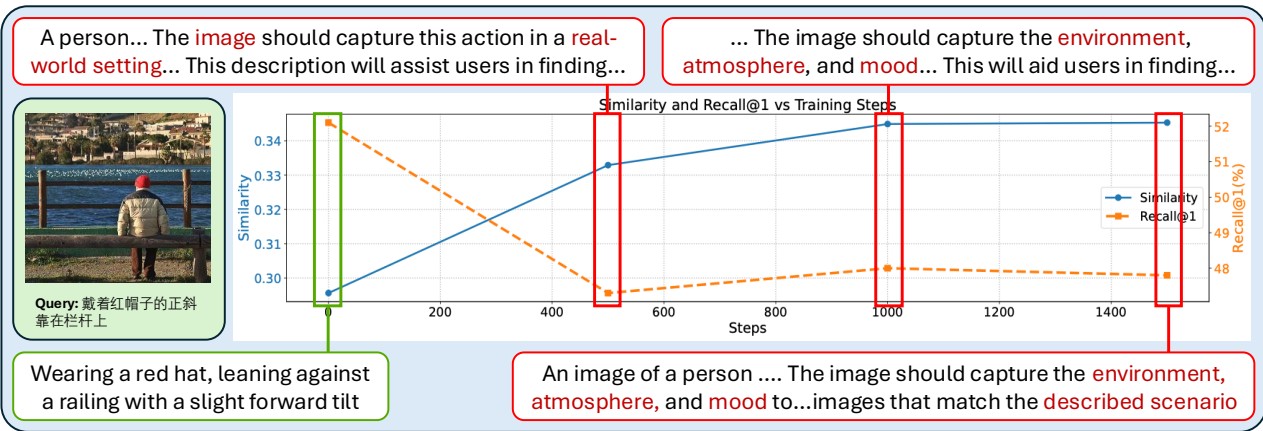

*Figure 5.* Using similarity scores as the reward leads to score inflation. As training epochs increase, the similarity between the query and the target image improves, but Recall@1 steadily decreases. The boxed text shown in the figure corresponds to the rewritten queries generated at the indicated training steps. Due to the lack of supervision signals, the LLM tends to generate generic or irrelevant words that raise similarity scores without improving retrieval quality.

While adapting queries without remapping the underlying embedding space offers significant practical flexibility, it also establishes a performance ceiling fundamentally dictated by the retriever's representational capacity. If the retriever lacks the requisite discriminative power for specific tasks, GRAPE can narrow the distribution gap but remains bounded by the retriever's inherent limits. Nonetheless, our empirical findings demonstrate that as the retriever's capability scales, GRAPE yields consistent and significant performance improvements.

**Inference and Efficiency Constraints.** Since every query must undergo LLM-based rewriting prior to retrieval, our framework inherently introduces additional inference latency. Although this overhead is manageable within academic benchmarks, deploying GRAPE in high-throughput industrial systems presents non-trivial practical challenges. Optimizing the trade-off between retrieval precision and computational cost—potentially through model distillation or lightweight architecture design—remains a pivotal direction for future research.

## 6. Conclusion

This work addresses the critical challenge of enhancing CLIP-based retrieval systems under distributional shifts without the prohibitive costs of retriever retraining or corpus re-embedding. We present **GRAPE**, a plug-and-play framework that leverages LLM-based query rewriting guided by fine-grained retrieval feedback. A key technical contribution is our corpus-relative ranking-based reward, which explicitly aligns policy optimization with ranking objectives and effectively obviates *score inflation*—a common pitfall in similarity-based supervision where irrelevant candidates receive indiscriminately high scores. Empirical evaluations across five representative benchmarks demonstrate that GRAPE consistently improves retrieval performance across multilingual, long-form, and multimodal shifts, yielding an average improvement of $4.9\%$ in Recall@10. These results underscore the efficacy of ranking-aware supervision in bridging distributional gaps while maintaining seamless compatibility with frozen retrievers. Our findings provide a promising pathway for developing scalable, adaptable

retrieval enhancements that remain decoupled from the underlying embedding infrastructure.

## Acknowledgements

The authors would like to thank the anonymous reviewers and area chairs for their constructive feedback and insightful suggestions. We sincerely thank Tencent Hunyuan AI Data for their invaluable support, including financial funding, computational resources, and the research environment that made this work possible. This work was also supported in part by the National Natural Science Foundation of China (Grant No. 62476041) under the project "Research on Multimodal Low-Level Vision Problems Based on Physical Processes and Object-Oriented Knowledge Architecture". Special thanks are extended to Zhixun Su and Jihong Zhang for their continuous mentorship, support, and guidance throughout this work.

## Impact Statement

This work introduces GRAPE to enhance information retrieval across diverse languages and modalities without requiring expensive model retraining or corpus re-embedding. By lowering the computational and economic barriers to high-quality retrieval, our approach fosters more inclusive access to global knowledge and promotes sustainable AI practices through improved data efficiency. However, as GRAPE relies on Large Language Models (LLMs) for query rewriting, it may inadvertently inherit or amplify biases present in pre-trained models. We encourage practitioners to conduct ethical auditing when deploying this framework in sensitive domains to ensure the fairness and neutrality of rewritten queries while maintaining robust retrieval alignment.

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

# A. Appendix

## A.1. Analysis of the Ranking-based Reward Function

In this section, we provide a comprehensive theoretical and empirical analysis demonstrating why the ranking-based reward formulation in GRAPE is fundamentally better suited for query rewriting than optimizing absolute similarity scores.

**Score Inflation under Absolute Similarity.** Let an image feature be decomposed as

$$z = \mu + \tilde{z},$$

where $\mu$ is the shared corpus mean and $\tilde{z}$ is the sample-specific component. Optimizing absolute similarity gives

$$q^\top z = q^\top \mu + q^\top \tilde{z}.$$

This objective inadvertently incentivizes the rewriter to generate generic tokens that increase the shared term $q^\top \mu$, uniformly inflating similarity scores across candidates without actually improving retrieval accuracy. In contrast, retrieval quality strictly depends on relative ranking:

$$q^\top z^+ > q^\top z^- \iff q^\top(\mu + \tilde{z}^+) > q^\top(\mu + \tilde{z}^-) \iff q^\top(\tilde{z}^+ - \tilde{z}^-) > 0.$$

Because the shared mean $\mu$ cancels out, optimizing for ranking encourages the model to focus on target-specific discriminative features rather than globally inflating similarity scores.

**Ranking-to-Reward Mapping.** Given a ranking $r_k \in \{1, \ldots, N\}$, consider a general affine mapping:

$$R_k = ar_k + b, \quad a \neq 0.$$

Our specific reward $R_k^r = 1 - \frac{2(r_k - 1)}{N-1}$ represents a special case of this mapping.

**Affine Invariance of Relative Advantage.** For a query $q$ with $K$ rewrites $\{R_k\}_{k=1}^K$, GRPO computes the advantage as:

$$\tilde{A}_k = \frac{R_k - \mu_q}{\sigma_q}, \quad \mu_q = \frac{1}{K} \sum_{j=1}^K R_j, \quad \sigma_q^2 = \frac{1}{K} \sum_{j=1}^K (R_j - \mu_q)^2.$$

Substituting $R_k = ar_k + b$ yields:

$$\mu_q = a\mu_r + b, \qquad \sigma_q = |a|\sigma_r,$$

where $\mu_r$ and $\sigma_r$ are the mean and standard deviation of $\{r_j\}$. Consequently,

$$\tilde{A}_k = \frac{a(r_k - \mu_r)}{|a|\sigma_r} = \text{sgn}(a) \cdot \frac{r_k - \mu_r}{\sigma_r}.$$

The normalized advantage remains invariant under affine transformations of the reward, up to a global sign flip. This property ensures that the optimization trajectory is dictated exclusively by the relative ordering of candidates, rendering it robust to arbitrary score shifts or scaling.

**Variance Stability.** During homogeneous exploration phases, continuous similarity scores often collapse into a narrow range, causing the reward variance to approach zero and pathologically amplifying gradient noise. By contrast, ranking rewards provide discrete ordinal signals. Even minute rank differences among rewrites yield a stable relative signal, substantially mitigating the risk of numerical instability during GRPO normalization.

**Empirical Comparison with Similarity-based Rewards.** To empirically validate the above analysis, we track the progression of ranking-based and similarity-based rewards during training. As illustrated in Figure 6, similarity-based rewards suffer from severe score inflation: the average similarity inflates while the actual Recall@1 steadily degrades. Conversely, ranking-based rewards yield stable and continuous improvements in retrieval recall, demonstrating that our proposed reward mechanism offers both theoretical robustness and practical training stability.

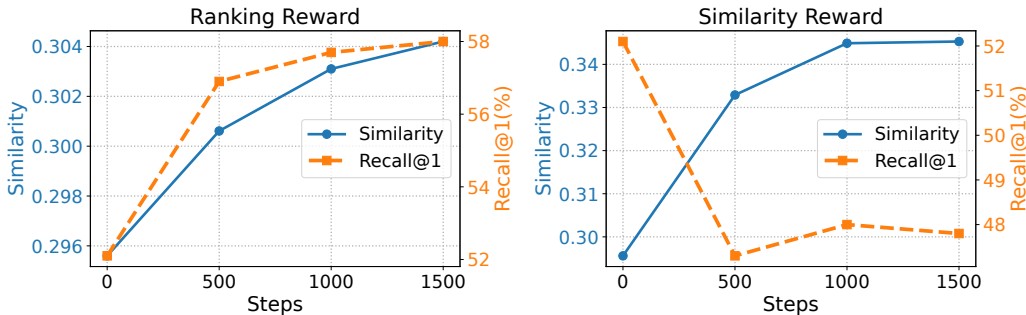

*Figure 6.* Comparison of similarity-based and ranking-based rewards during training. Ranking-based rewards effectively prevent score inflation and align seamlessly with improvements in retrieval recall.

*Table 2.* Training and validation statistics of the datasets used in our experiments.

| Dataset | Train Queries | Train Images | Val Queries | Val Images |
|---|---|---|---|---|
| Flickr30k-CN | 148,915 | 29,783 | 5,000 | 1,000 |
| CVLUE | 89,600 | 17,920 | 15,580 | 3,116 |
| Wikipedia | 2,173 | 2,173 | 693 | 693 |
| XM3600 | 126,341 | 1,745 | 33,218 | 456 |
| CIRR | 3,336 | 1,951 | 834 | 723 |

## A.2. Experimental Setup Details

**Datasets.** Table 2 details the training and validation splits for all datasets utilized in our experiments, outlining the specific volume of images and queries per split.

**Prompt Templates.** As shown in Table 3,we engineered task-specific prompts tailored for multilingual retrieval, multimodal fusion, and long-form text queries. These templates provide structural constraints, guiding the LLM rewriter to generate retrieval-optimal inputs for the CLIP encoder.

## A.3. Additional Training-free Baseline Comparisons

To isolate the source of GRAPE's performance gains and verify that they stem from retriever-aligned optimization rather than the inherent capacity of the underlying language model or prompt engineering, we benchmark GRAPE against advanced training-free rewriting baselines, including PlugIR and state-of-the-art LLMs. For PlugIR, we exclusively utilize its query rewriting component without iterative interactive feedback, ensuring a rigorous and controlled comparison within the training-free paradigm.

As shown in Table 4,GRAPE demonstrates significant advantages over PlugIR and remains highly competitive with much larger proprietary LLMs (e.g., Claude-Sonnet-4.6 and GPT-5.4) evaluated under the exact same prompt settings. Without explicit retriever supervision, PlugIR and these zero-shot LLM baselines tend to generate queries that are semantically plausible but poorly aligned with the retriever's embedding space. Conversely, GRAPE leverages direct ranking-based supervision from the retriever, tightly aligning the semantic structure of the rewrites with the retriever's latent geometry. Impressively, despite employing a relatively compact backbone (Qwen2.5-3B), GRAPE surpasses GPT-5.4 across all metrics and outperforms Claude-Sonnet-4.6 in Recall@1, remaining comparable in Recall@10 on Flickr-CN. This confirms that the efficacy of our framework originates from target-aware optimization rather than mere parameter scaling.

## A.4. Long-form Query Characterization

We conduct a granular analysis on the Wikipedia dataset to explicitly characterize the dynamics of long-form query processing. In Table 5, the "LLM Explicit Shorten" variant represents a controlled setting where the LLM is strictly prompted to yield highly compressed rewrites.

GRAPE synthesizes significantly more concise and stable queries, effectively bridging the distribution gap with the retriever's

*Table 3.* Prompt templates utilized across different retrieval scenarios.

| Task | Prompt Template |
|------|-----------------|
| Multilingual | `You're an image retrieval assistant. Translate search queries: {text} into optimized English text for vector-based image search. Show your work in <think></think> tags. And return the final text in <answer></answer> tags.` |
| Multimodal | `You're an image retrieval assistant. You need to use the information in the image, generate a text based on the description in the text: {text}. This generated text will be used by CLIP to retrieve the corresponding image. Show your work in <think></think> tags. And return the final text in <answer></answer> tags.` |
| Length | `You are an image retrieval assistant. Summarize the given Wikipedia text content: {text} into a concise visual description suitable for CLIP model input to retrieve corresponding images of the described subject. Show your work in <think></think> tags. And return the final text in <answer></answer> tags.` |

*Table 4.* Comparison with advanced training-free rewriting baselines. Metrics reported are Recall@1 / Recall@10 (%).

| Dataset | Claude-Sonnet-4.6 | GPT-5.4 | PlugIR | **GRAPE** |
|---------|-------------------|---------|--------|-----------|
| Flickr-CN (1k) | 56.2 / 91.3 | 56.2 / 89.0 | 51.4 / 84.4 | **58.7 / 90.1** |
| Wikipedia | 36.9 / 73.9 | 38.4 / 78.4 | 33.2 / 73.0 | **43.7 / 82.8** |

optimal input space. Importantly, length reduction alone is insufficient to explain the performance surge: artificially forcing the LLM to shorten outputs to an average of 35.7 words yields results vastly inferior to GRAPE. This underlines that GRAPE's mechanism extends beyond mere compression; it acts as a selective filter that distills vital visual entities and preserves structural semantics tailored specifically for the retrieval objective.

## A.5. Robustness to Group Size

The GRAPE framework exhibits strong stability across moderate group sizes. We adopt $K = 8$ as our default configuration, as it strikes an optimal balance by providing ample reward diversity without excessive computational overhead. Table 6 delineates the impact of varying the parameter $K$.

The performance is remarkably robust across varying group sizes, exhibiting a mild optimum at $K = 8$. This stability inherently stems from the discrete nature of ranking rewards, which are significantly less susceptible to the marginal variance fluctuations that typically plague continuous similarity-based rewards. The presence of even slight ordinal differences among the generated group provides a sufficiently strong supervisory gradient.

## A.6. Failure Mode Analysis

As GRAPE builds upon a base LLM rewriter, it inherits initial susceptibilities to typical generative failure modes, such as hallucination or linguistically biased extrapolations. However, the retriever-in-the-loop feedback mechanism effectively acts as a natural regularizer against these pathologies. Because the reward signal is strictly dictated by the ultimate retrieval ranking, rewrites that inject spurious, deceptive, or structurally malformed tokens inevitably degrade the ranking metric and are consequently heavily penalized during the GRPO update phase. In essence, the retriever functions as an uncompromising, implicit discriminator of rewrite fidelity.

*Table 5.* Length statistics and retrieval performance on the Wikipedia dataset. Metrics reported are Recall@1 / Recall@10 (%).

| Method | Avg. Length | Std. Dev. | Recall@1 / Recall@10 |
|---|---|---|---|
| Raw Query | 533 | 237 | 33.5 / 71.9 |
| LLM Rewrite | 91 | 45 | 34.3 / 69.3 |
| LLM Explicit Shorten | 35.7 | 13 | 34.8 / 73.7 |
| **GRAPE** | **45** | **14** | **43.7 / 82.8** |

*Table 6.* Sensitivity analysis of group size $K$ on Flickr30k-CN. Metrics reported are Recall@1 / Recall@10 (%).

| Number of Rewrites | Recall@1 / Recall@10 |
|---|---|
| $K = 2$ | 57.1 / 89.2 |
| $K = 4$ | 57.3 / 89.4 |
| $K = 8$ | **58.0 / 90.0** |

To quantitatively assess this self-correcting behavior, we conducted a manual audit of 100 randomly sampled queries from the Flickr30k dataset. The baseline, unoptimized LLM exhibited a 14% failure rate, characterized by distinct issues such as semantic concept stacking, catastrophic generation loops (e.g., repeating hashtags), and prompt-following failures (e.g., outputting mixed Chinese-English syntax). Post-optimization, GRAPE dramatically suppressed this failure rate to a mere 3%. The residual errors were predominantly cases of benign over-expansion, which only marginally dilute the exact precision and rarely trigger catastrophic retrieval failures.

### A.7. Additional Discussion and Practical Guidelines

**When GRAPE Helps Most and Its Inherent Bottlenecks.** GRAPE demonstrates the most pronounced improvements under substantial distribution shifts between user queries and the retriever's original training data. We identify three primary scenarios where this holds true. First, in instances of severe length shift, such as long-form Wikipedia queries, GRAPE excels at semantic distillation and redundancy elimination. Second, when confronting semantic or compositional complexity, as seen in the CIRR dataset, GRAPE successfully re-contextualizes and extracts retrieval-critical attributes. Third, in cases of cross-lingual mismatch, such as Flickr30k-CN, GRAPE goes beyond literal translation to perform deep semantic enrichment.

Conversely, the efficacy of GRAPE is bounded when the fundamental bottleneck lies in absolute model capacity rather than query formulation. If the frozen dual-encoder entirely lacks representational grounding for a given concept, or if the backbone LLM possesses negligible knowledge for a specific domain, optimization via query rewriting cannot circumvent these hard capacity limits. This limitation is empirically evident in our multilingual evaluation. As illustrated in Figure 7, we provide a comprehensive visualization of per-language retrieval performance on the XM3600 dataset. These radar plots underscore the dichotomy between high-resource and low-resource languages: GRAPE acts as a powerful amplifier for languages where the LLM already possesses robust internal representations (e.g., English, Chinese, Spanish), but the performance delta narrows considerably for statistically underrepresented languages. This behavior reinforces the premise that GRAPE's downstream task versatility is inherently anchored to the pre-existing knowledge taxonomy of the backbone LLM. Nonetheless, as foundational models continue to scale in multilingual proficiency, the operational envelope of the GRAPE framework will natively expand in tandem.

**Efficiency and Latency.** During deployment, the GRAPE pipeline necessitates a single forward pass through the LLM for query reformulation, immediately followed by the standard vector search. The time complexity is thus $O(T_{\text{LLM}} + T_{\text{retrieval}})$. While this naturally introduces a latency overhead compared to zero-shot, unrewritten retrieval, it remains highly competitive with other state-of-the-art rewriting methodologies. This is primarily because GRAPE's optimization fundamentally enhances the rewriting efficiency of smaller parameter models, allowing for the deployment of compact LLMs (such as Qwen2.5-3B) coupled with streamlined system prompts. As a result, this approach significantly reduces computational overhead and effectively alleviates the latency issues commonly encountered when integrating LLMs into retrieval systems.

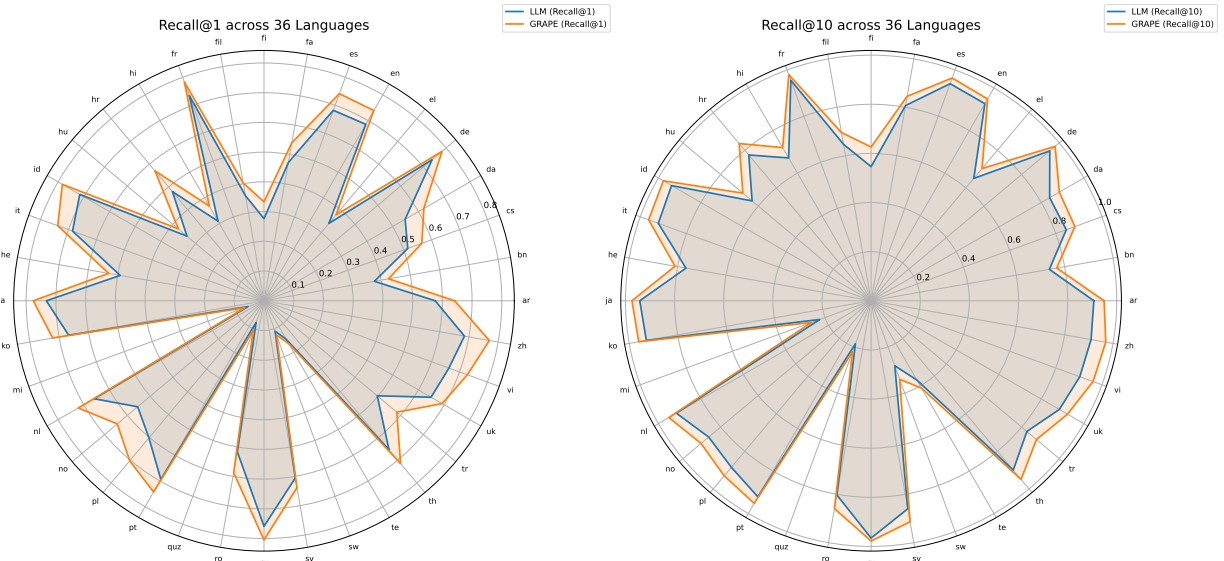

*Figure 7.* Radar plots illustrating Recall@1 (top) and Recall@10 (bottom) for GRAPE on the XM3600 dataset across 36 languages. The performance variance highlights the correlation between LLM intrinsic knowledge coverage and GRAPE's optimization ceiling.

