# OpenReview forum: "GRAPE: Let GRPO Supervise Query Rewriting by Ranking for Retrieval"
_ICML.cc/2026/Conference — ICML 2026 regular_

### Official Review · Reviewer_TjVA · 2026-03-05

**Soundness:** 3
**Presentation:** 3
**Significance:** 3
**Originality:** 2
**Overall Recommendation:** 3
**Confidence:** 4

**Summary:**

This paper introduces GRAPE, a plug-and-play framework that improves CLIP-based retrieval under distributional shifts (multilingual, long-form, multimodal) by training an LLM query rewriter via GRPO with ranking-based rewards. A key contribution is identifying score inflation in similarity-based supervision and addressing it with a corpus-relative ranking reward, achieving an average 4.9% gain in Recall@10 across five benchmarks without modifying the frozen retriever.

**Compliance With Llm Reviewing Policy:**

Affirmed.

**Final Justification:**

I have read the rebuttal; Concerning of the novelty, i will hold the point, but admits the work's solid performance

**Key Questions For Authors:**

Given that multilingual, long-form, and multimodal retrieval are fundamentally different challenges with dedicated prior work, why does GRAPE apply the same GRPO framework uniformly across all three rather than incorporating task-specific designs?

Further, I also wonder whether more powerful retrieval models suffer from the same issues, such as GME or Qwen3-VL-Embedding. It remains unclear whether this phenomenon is specific to CLIP or exists more generally across embedding models. Additional experiments would help verify this and make the conclusion more convincing. For now, this work essentially amounts to query rewriting. While it is effective, its performance stays within expectations.

**Limitations:**

yes

**Strengths And Weaknesses:**

## **Pros**
**Well motivated:** Score inflation is a concrete and previously overlooked failure mode. The paper backs it up with both a theoretical proof and clear experiments showing similarity scores going up while recall goes down.

**Simple and plug-and-play:** The method requires no changes to the retriever or re-embedding of the corpus, and works consistently across multiple languages, query types, and CLIP model sizes.

## **Cons**
**Limited novelty:** Using LLM-based query rewriting to improve CLIP retrieval is well-established, and addressing score inflation via ranking-based rewards within GRPO does not offer genuinely new insights. The method is effective but largely an engineering combination of existing ideas.

**Weak competitive positioning:** GRAPE is only compared against narrow baselines. Universal retrieval models like GME[1], if simply paired with an LLM translator, would likely match or exceed GRAPE's performance with much broader generality, making the practical advantage of GRAPE unclear.

[1] GME: Improving Universal Multimodal Retrieval by Multimodal LLMs

---

> ### Author Rebuttal · Authors · 2026-03-30
>
> We thank the reviewer for the insightful comments and constructive questions.
>
> **Q1. Why a unified GRPO framework across different settings?**
>
> While multilingual, long-form, and multimodal retrieval differ in surface form, they share a common underlying challenge: distribution mismatch between query inputs and the retriever’s latent space.
> GRAPE is designed to address this unified learning problem, rather than introducing task-specific heuristics. The key idea is that the supervision signal—retrieval ranking—is task-agnostic but retriever-specific, and directly reflects retrieval quality across all settings.
> As a result, the same GRPO framework naturally induces different rewriting behaviors depending on the distribution:
> - translation and semantic enrichment for multilingual queries,
> - distillation and compression for long-form queries,
> - cross-modal fusion for multimodal queries.
>
> This behavior is empirically demonstrated in Sec. 4.3 of our paper. Importantly, this unified design avoids task-specific engineering while maintaining strong performance across diverse settings, which is a key practical advantage.
>
> **Q2. Competitive positioning and comparison with stronger backbones (e.g., GME)**
>
> We agree that stronger retrievers such as GME are important for evaluating robustness. we have conducted additional experiments using a much stronger and more recent backbone model, **GME-2B**.The results are summarized in the table below (Metrics: Recall@1 / Recall@10 %):
>
> | Dataset | GME-2B (Baseline) | GME-2B + LLM | **GME-2B + GRAPE (Ours)** |
> | :--- | :---: | :---: | :---: |
> | Flickr-CN | 68.86 / 94.24 | 69.7 / 94.1 | **72.0 / 94.6** |
> | Wikipedia | 38.67 / 73.6 | 36.9 / 70.7 | **40.2 / 74.2** |
> | CIRR | 32.49 / 77.7 | 44.2 / 85.3 | **46.2 / 87.5** |
>
>
> Notably, the reason GME performs well on Flickr-CN is that its training data included Chinese. Despite this high baseline, these results suggest that GRAPE is complementary rather than replaceable: it improves retrieval by aligning queries with the retriever’s latent space, regardless of the backbone strength. In contrast, simply pairing a stronger retriever with prompt-based rewriting does not explicitly align query rewriting with retrieval quality, as it lacks direct feedback from the retriever.
> We will include these results and clarify the positioning in the revised version.
>
> **Q3. Novelty and contribution beyond existing rewriting approaches**
>
> We clarify that GRAPE is not simply a combination of existing components, but introduces a tailored formulation of query rewriting as a retriever-aligned optimization problem for cross-modal spaces. While most prior work treats rewriting primarily as prompt engineering, GRAPE specifically addresses the unique distributional shifts (multilingual, long-form, and multimodal) inherent to frozen vision-language models like CLIP. Our novelty extends beyond the use of ranking feedback: we explicitly identify the pathological 'score inflation' phenomenon caused by similarity-based finetuning—where irrelevant candidates receive indiscriminately high scores—and demonstrate how our corpus-relative ranking reward effectively mitigates this degenerate behavior.

---

> > ### Author Rebuttal · Reviewer_TjVA · 2026-04-01
> >
> > Thank you for the rebuttal and the additional GME-2B experiments.
> >
> > However, the core idea of using ranking feedback from a retriever to supervise query rewriting, rather than pointwise similarity, is not new. RaFe[1] published over a year before GRAPE, proposes precisely this. The superiority of ranking-based over pointwise supervision is also well-established in the Information Retrival literature more broadly, and applying GRPO with retrieval ranking signals is now a standard practice.
> >
> > GRAPE's actual contribution is instantiating this known principle within the CLIP+LLM pipeline and empirically identifying score inflation in that setting. This is a valid finding, but not a conceptually new one for ICML, I believe it is better positioned for a venue like CVPR or ECCV, where engineering contributions with strong empirical validation are valued on their own terms.
> >
> > [1]RaFe: Ranking Feedback Improves Query Rewriting for RAG

---

> > > ### Author Response · Authors · 2026-04-07
> > >
> > > We sincerely thank you for your insightful feedback.
> > >
> > > Here, we hope to provide a clearer perspective from two dimensions: the essential differences between our work and frameworks like RaFe, and theoretical proofs of our design. These illustrate why GRAPE is not a simple cross-modal engineering port, but rather a method grounded in optimization analysis, designed to overcome the numerical challenges of applying RL in cross-modal retrival:
> > >
> > > **1. Pure Text Local Score Comparison vs. Cross-Modal Global Target Rank Anchoring**
> > >
> > > RaFe's "ranking feedback" essentially utilizes a re-ranker to compute absolute relevance scores between the original query and the retrieved Top-K documents one by one, taking the average as the reward signal. In essence, this remains a local aggregated optimization based on absolute similarity scores, rather than a true global ranking signal. In contrast, GRAPE discards absolute similarity scoring. Instead, it directly utilizes the global ranking of the target image within the retrieval results as the sole supervision signal. This enables the rewriting model to adapt to the pre-trained latent space of the frozen retriever, thereby effectively mitigating retrieval performance degradation caused by distribution shifts and achieving alignment between query rewriting optimization and cross-modal recall performance.
> > >
> > > **2. Theoretical Proofs: Mathematical Guarantees for RL Optimization**
> > >
> > > Addressing the empirical limitations of reward designs in existing methods, we demonstrate three theoretical results explaining why discrete global ranking signals are more robust than continuous similarity signals:
> > >
> > > * **Revealing the mathematical root of "Score Inflation":** We can decompose the rewritten result via vector decomposition into a mean direction and a specificity direction. Optimizing directly toward absolute similarity induces the model to find a shortcut—generating generic tokens lacking specificity to indiscriminately inflate global scores, ultimately causing the query representation to irreversibly collapse toward the corpus mean.
> > > * **Anchoring relative advantage via affine invariance:** To resolve the aforementioned issue, we propose a linear reward mechanism based on discrete rankings, $R_k = a \cdot r_k + b$. Thanks to the affine invariance of GRPO's group normalization, the advantage function computation can be strictly simplified to:
> > >   $$\tilde{A}_k = \frac{(a \cdot r_k + b) - (a \cdot \mu_r + b)}{|a| \cdot \sigma_r} = \text{sgn}(a) \cdot \frac{r_k - \mu_r}{\sigma_r}$$
> > >   This derivation mathematically establishes that GRAPE's optimization gradient is completely and exclusively anchored to the relative ordinal relationships among samples. This forces the model to focus on improving true global rankings, mechanistically immunizing it against score shifts and inflation.
> > > * **Establishing a strictly positive lower bound for discrete variance to eliminate numerical pathology:** During the homogeneous exploration phase, continuous scores cause the reward variance to approach zero ($\sigma_q \to 0$), triggering gradient amplification and training oscillation. The discrete rankings ($r_k \in \mathbb{Z}^+$) introduced by GRAPE fundamentally eliminate the vanishing denominator problem. For example, under the $K=8$ setting, even if the intra-group rewrites are extremely similar (with only a 1-rank difference), the variance maintains a strictly positive minimum value (standard deviation $\sigma_{\min} = 0.331$). This "variance floor" theoretically circumvents the risk of numerical collapse in RL training.

---

### Official Review · Reviewer_91p6 · 2026-03-11

**Soundness:** 4
**Presentation:** 4
**Significance:** 4
**Originality:** 4
**Overall Recommendation:** 5
**Confidence:** 4

**Summary:**

GRAPE improves CLIP-based retrieval when queries don’t match the data that CLIP was trained on (e.g. image+text). Instead of retraining CLIP or embedding the corpus again, it keeps the retriever fixed and trains an LLM to rewrite queries so they align better with CLIP’s embeddings.
The main idea is to supervise this rewriter with ranking from the fixed retriever (how high the target item is ranked) rather than with raw similarity scores. The authors show that optimizing for similarity leads to score inflation, they look good by similarity but don’t improve retrieval, and they use Group-Relative Policy Optimization (GRPO) so the LLM gets a reward based on actual ranking quality. In experiments on multilingual, long-form, and multimodal benchmarks, GRAPE increases the recall (e.g. ~4.9% average in R@10, the fraction of queries where the correct item appears anywhere in the top‑10 results returned by the retriever.) without changing the retriever or the index.
Its key contributions are:
A method (GRAPE) that keeps the retriever fixed and improves retrieval by optimizing LLM query rewriting with retrieval feedback, without any need to retrain the retriever or embed the corpus again.
Showing that similarity-based supervision causes ``score inflation” and introducing a ranking-based rewarding system to align with the actual ranking quality and yields rewrites more specific to the target.
Showing consistent gains on benchmarks (multilingual, long-form, multimodal), with an average +4.9% Recall@10.
Conclusion:
GRAPE improves retrieval by training an LLM to rewrite queries using ranking feedback from a fixed CLIP retriever instead of similarity scores, giving consistent gains without retraining the retriever or re-embedding the corpus. Using a ranking-based rewarding system avoids score inflation and results in rewrites more specific to the target that actually improve recall.

**Compliance With Llm Reviewing Policy:**

Affirmed.

**Key Questions For Authors:**

1) When the number of rewrites per query (K) is small, the spread of rewards can be near zero, so the normalized advantage may be unstable. Do you have safeguards for small (K)?
      a) How responses would change our evaluation: Evidence of stability for small (K) or an explicit choice of (K) would address our        soundness concern; if left unaddressed, we would keep that as a limitation.

2) The system’s behavior basically depends on the rewriter. Have you considered how typical LLM failure modes (e.g., hallucination, biased or stereotyped rewrites) affect retrieval quality or fairness?
      a) How responses would change our evaluation: Any analysis would strengthen our assessment of soundness and significance; if not considered, we would continue to treat this as a limitation.

**Limitations:**

Yes. The authors discuss limitations (e.g. LLM knowledge bottleneck, retriever-dependent cap, inference and efficiency constraints) and societal impact (e.g. bias in rewrites, need for ethical auditing in sensitive domains). A short note on how LLM failure modes (e.g., hallucination, bias) might affect rewriter behavior in production would further strengthen the limitations section, but overall it is well-explained.

**Strengths And Weaknesses:**

Soundness
Overall, the submission seems technically sound: the method is well-specified, the rewarding system has a clear theoretical ground, the score inflation failure is demonstrated and addressed practically, and limitations are stated.
Clear problem framing and formulation.
Sensible rewarding system design with theoretical grounding. The link between reward and retrieval metric (rank of target) makes sense.
Practical validation of ``score inflation”.
Addressed limited knowledge for languages with lower resources and inference cost as bottlenecks as a result of using LLMs.
Whether rank is over full corpus or a subset for large (N) is not fully clarified.

Presentation :
Presentation looks strong; structure, figures, and reproducibility details are well-explained.
Narrative is clear (distribution shift → query adaptation → need for auditing → score inflation → rewarding system).
Structured related work (training-based and training-free).
Reproducible.

Significance :
The paper addresses an important practical problem and shows consistent gains across several shift types, provides data-efficient training, and offers a new insight (ranking vs similarity rewarding system).
The paper addresses the important problem of the setting where re-embedding and retraining are not feasible.
Evaluations cover scenarios with various multilingual, size, and modality setups.
The score-inflation argument is broadly relevant; optimizing a pointwise score like similarity score can diverge from the real ranking objective because of its limitations.
The evidence that with 10% of training data, GRAPE recovers at least 88% of full-data Recall@1 is relevant for domains where labeled retrieval data is rare and supports the claim that the rewriter is learning ``retriever preferences” from ranking feedback rather than memorizing large supervised datasets.
The authors acknowledge inclusivity and sustainability (lower compute, no re-embedding) and risks (LLM bias in rewrites, need for auditing in sensitive domains), which aligns significance with responsible deployment.
Impact is broad in terms of shift type (language, length, modality).
The scope is only specialized in that it assumes a fixed CLIP-style retriever and LLM-based rewriting.

Originality:
A combination of existing methods with a clear new insight (score inflation and similarity vs ranking) and well-justified design; originality is in framing the problem, ranking reward, and analysis.
GRPO is for retrieval-guided query rewriting with a ranking reward and fixed retriever.
Score inflation is identified and tied to pointwise similarity; the ranking-based fix is motivated and tested.
The evaluation analysis covers different shift types. The rewriter is trained with retrieval feedback while the retriever stays training-free.
Building blocks (GRPO, LLM rewriting, CLIP, etc.) are existing; novelty is actually in combination, reward design, and analysis rather than wholly new machinery.

---

> ### Author Rebuttal · Authors · 2026-03-30
>
> We thank the reviewer for the positive and insightful feedback.
>
> **Q1. Stability when K is small**
>
> Empirically, we find GRAPE to be stable with moderate group sizes. In our experiments, we use K=8, which provides sufficient reward diversity. We additionally evaluate different K values:
>
> | Number of Rewrites (K) | Recall@1 / Recall@10 (%) |
> | :--- | :---: |
> | K=2  | 57.1 / 89.2 |
> | K=4 | 57.3 / 89.4 |
> | **K=8 (Our Default)** | **58.0 / 90.0** |
> | K=16 | 57.7 / 89.7 |
>
> Performance is relatively robust, with a mild optimum around K=8. We also observe stable training without collapse(see Fig. 6 in the paper).
>
> Intuitively, GRAPE is less sensitive to small reward variance compared to similarity-based rewards. Since our reward is derived from relative rankings, even minimal rank differences between rewrites lead to consistent ordinal signals, avoiding the noise amplification issue commonly observed with dense similarity scores.
>
> We will clarify the choice of K and include this analysis in the final version.
>
> **Q2. Impact of LLM failure modes**
>
> We agree that GRAPE depends on the base rewriter and can be initially affected by typical LLM failure modes (e.g., hallucination or biased rewrites). However, GRAPE significantly mitigates these issues through its retriever-based feedback mechanism. Since the reward is directly derived from ranking performance, rewrites that introduce irrelevant, misleading, or poorly formatted content naturally degrade the retrieval ranking and are therefore strictly penalized during optimization. In this sense, the retriever acts as a robust, implicit discriminator over rewrite quality.
>
> To empirically quantify this mitigation, we conducted a manual evaluation on a randomly sampled subset of 100 queries from the Flickr30k dataset. The analysis revealed a stark contrast in generation robustness:
>
> - Baseline LLM (14% failure rate, 14/100): The unoptimized baseline frequently exhibited severe failure modes, including concept stacking, runaway generation (e.g., repetitive hashtag loops), and instruction-following failures (such as outputting mixed Chinese-English queries instead of the target language). These errors severely disrupt the retrieval process.
> - GRAPE (3% failure rate, 3/100): GRAPE not only reduced the overall failure rate to 3%, but also restricted the severity of the errors. The only observed failure mode in the GRAPE-trained model was occasional "over-expansion" (generating slightly more detail than necessary). Crucially, this over-expansion only marginally affects the exact precision but does not lead to complete retrieval failure.
>
> We will explicitly include these manual evaluation findings and expand our discussion on failure modes in the revised paper.

---

> > ### Author Rebuttal · Reviewer_91p6 · 2026-04-04
> >
> > No Concern. My original recommendation stand.

---

> > > ### Author Response · Authors · 2026-04-07
> > >
> > > Sincere thanks for your careful review of our rebuttal, for confirming no further concerns, and for upholding your original recommendation. We greatly appreciate your time and professional input, and will carefully refine the final manuscript accordingly.

---

### Official Review · Reviewer_R1GA · 2026-03-12

**Soundness:** 2
**Presentation:** 3
**Significance:** 3
**Originality:** 2
**Overall Recommendation:** 3
**Confidence:** 4

**Summary:**

This paper studies how to improve CLIP-based retrieval under query-side distribution shifts, including multilingual, long-form, and multimodal queries, without retraining the retriever or re-embedding the corpus. The proposed method, GRAPE, uses an LLM to rewrite queries and optimizes the rewriter with GRPO using a ranking-based reward derived from the frozen retriever, motivated by the observation that similarity-based supervision can lead to score inflation and weak alignment with actual retrieval quality. Experiments on five benchmarks and three frozen CLIP backbones show consistent improvements over vanilla CLIP and frozen-LLM rewriting baselines, suggesting that retrieval-aware query rewriting is a practical way to adapt frozen retrieval systems.

**Compliance With Llm Reviewing Policy:**

Affirmed.

**Key Questions For Authors:**

- Given that query rewriting for retrieval is already a crowded design space, can the authors more clearly articulate what is uniquely advantageous about GRAPE, either through stronger theoretical justification or through more targeted comparisons with existing rewriting approaches?

- Can the authors provide a clearer efficiency analysis, including the time complexity of the proposed method and empirical inference latency measurements, especially given that retrieval is often a latency-sensitive setting in practice?

- How robust are the reported gains across different backbone models, and can the authors include results on at least one stronger and more recent backbone to better support the claimed generality of the framework?

**Limitations:**

See above

**Strengths And Weaknesses:**

Strengths

1. The paper is well written and easy to follow. The motivation, method description, and empirical setup are generally clear.

2. The paper studies a practical problem: adapting frozen CLIP-based retrieval systems to multilingual, long-form, and multimodal queries without retraining the retriever or re-embedding the corpus.

3. The ranking-based supervision is reasonably motivated, and the discussion of score inflation is potentially useful for understanding why pointwise similarity supervision may be suboptimal.

4. The experiments cover multiple datasets and CLIP backbones, and the reported gains appear broadly consistent.


### Weakness

1. The main weakness is that query rewriting for retrieval is already a fairly crowded design space, so the paper needs to more clearly establish what is uniquely advantageous about GRAPE, either through stronger theoretical justification or more targeted experimental analysis.
2. The paper would be stronger with a clearer efficiency analysis, including time complexity and empirical inference latency. This is especially important in retrieval settings, where query-time latency is often a key practical constraint.
3. The evaluation relies on a relatively narrow set of backbone models. Adding at least one stronger and more recent backbone would make the empirical validation more convincing and help clarify whether the observed gains are robust across different models, rather than being tied to the particular backbones used in the paper.

---

> ### Author Rebuttal · Authors · 2026-03-30
>
> We thank the reviewer for the constructive feedback.
>
> **Q1. What is uniquely advantageous about GRAPE?**
>
> We clarify that GRAPE’s key novelty lies in introducing retriever-aligned supervision into query rewriting.
>
> First, we add comparisons with a stronger training-free baseline, PlugIR (ACL’24 oral). To ensure a controlled setting, we use only its query rewriting component. GRAPE consistently outperforms PlugIR: on Flickr-CN (1k samples), from 51.4 / 84.4 to 58.7 / 90.1; on Wikipedia, from 33.2 / 73.0 to 43.7 / 82.8.
>
> More importantly, existing rewriting methods are primarily prompt-driven and lack explicit supervision from the retriever. As a result, they often produce semantically plausible but retrieval-suboptimal queries, especially under complex inputs. For example, even with strong LLMs (e.g., GPT-5.4), prompt-based rewriting tends to generate fluent but retrieval-suboptimal queries, as it is not explicitly aligned with the retriever’s embedding space.
>
> In contrast, GRAPE directly optimizes rewriting with ranking-based feedback from the retriever, aligning rewritten queries with the retriever’s latent space. Notably, even with a relatively small rewriter (Qwen2.5-3B), GRAPE consistently outperforms these approaches. For example, on Wikipedia, GPT-based rewriting achieves 38.4 / 78.4, while GRAPE achieves 43.7 / 82.8.
>
> From a theoretical perspective, GRAPE overcomes score inflation. We provide a theoretical analysis below:
>
> * **Feature Decomposition:** Let an image feature $z$ in the latent space be decomposed into a shared corpus mean $\mu$ and a sample-specific vector $\tilde{z}$:
> $$z=\mu+\tilde{z}$$
> * **The Pitfall of Absolute Similarity:** Optimizing absolute similarity yields the objective:
> $$q^\top z=q^\top\mu+q^\top\tilde{z}$$
> This creates a severe optimization shortcut: the LLM generates generic words to maximize the shared term $q^\top\mu$, uniformly inflating scores across all candidates. Crucially, absolute similarity fails to provide the correct optimization direction, as its gradient blindly pushes the LLM toward the uninformative corpus mean $\mu$.
> * **The Advantage of Relative Ranking (Ours):** Conversely, actual retrieval relies on relative ranking, which depends strictly on the margin between positive ($z^+$) and negative ($z^-$) samples:
> $$q^\top z^+>q^\top z^-\iff q^\top(\mu+\tilde{z}^+)>q^\top(\mu+\tilde{z}^-)\iff q^\top(\tilde{z}^+-\tilde{z}^-)>0$$
> By directly optimizing the ranking reward, GRAPE mathematically cancels out the shared mean $\mu$. **This allows GRAPE to explicitly capture the exact direction for optimization:** forcing the LLM to focus entirely on distinguishing features ($\tilde{z}^+-\tilde{z}^-$) and guiding it precisely toward generating target-specific concepts.
>
>
>
> **Q2. Efficiency and Latency Analysis**
>
> We agree that efficiency is important in retrieval settings and will clarify this in the revision.
>
> At inference time, GRAPE introduces one additional LLM forward pass for query rewriting, followed by standard retrieval. The overall complexity is therefore O(T_LLM + T_retrieval).
>
> In practice, this introduces moderate latency overhead compared to vanilla retrieval, but remains comparable to existing query rewriting approaches. Importantly, GRAPE enables the use of smaller LLMs with simpler prompts due to its improved rewriting capability, which can partially offset the additional cost. We will include empirical latency measurements and a more detailed efficiency analysis in the revised version.
>
>
> **Q3. Robustness across backbones**
>
> To further demonstrate the robustness and plug-and-play generality of our framework, we have conducted additional experiments using a much stronger and more recent backbone model, **GME-2B**.The results are summarized in the table below (Metrics: Recall@1 / Recall@10 %):
>
> | Dataset | GME-2B (Baseline) | GME-2B + LLM | **GME-2B + GRAPE (Ours)** |
> | :--- | :---: | :---: | :---: |
> | Flickr-CN | 68.86 / 94.24 | 69.7 / 94.1 | **72.0 / 94.6** |
> | Wikipedia | 38.67 / 73.6 | 36.9 / 70.7 | **40.2 / 74.2** |
> | CIRR | 32.49 / 77.7 | 44.2 / 85.3 | **46.2 / 87.5** |
>
> As shown above, GRAPE consistently improves performance and achieves the best results even when paired with a highly capable modern backbone like GME-2B. Notably, GME-2B achieves strong baseline results on Flickr-CN because its training data included Chinese. Despite such an optimized baseline, GRAPE successfully pushes the performance ceiling even higher. In contrast, standard LLM rewriting sometimes struggles to improve upon such a strong backbone—and even degrades performance on Wikipedia—whereas GRAPE effectively aligns the rewriter with the specific embedding space of the retriever.
>
> This  validates that GRAPE's improvements are robust across different retrievers and that our framework can generalize as a plug-and-play enhancer for future state-of-the-art embedding models. We will include these experiments in the revised manuscript to further support the claimed generality.

---

> > ### Author Rebuttal · Reviewer_R1GA · 2026-04-03
> >
> > I think the novelty problem is still a concern.

---

> > > ### Author Response · Authors · 2026-04-07
> > >
> > > We sincerely thank you for your insightful feedback.
> > >
> > > Regarding the innovation of this study, we would like to further clarify: our work is by no means a simple engineering application, but rather makes substantial contributions to both the cross-modal retrieval paradigm and reinforcement learning optimization theory. Specifically, to address the distribution shift between user queries and the retriever's training data, to the best of our knowledge, we are the first in the cross-modal retrieval domain to propose a GRPO-based method that strictly uses "ranking" as the supervision signal, effectively mitigating this issue without fine-tuning the retriever. More importantly, during our research, we identified the inherent "score inflation" phenomenon in GRPO and provided a mathematical proof of its underlying mechanism via vector decomposition. Concurrently, we theoretically derived and proved the affine invariance and variance lower bound properties of the ranking reward function:
> > >
> > > **1. Affine Invariance:** Defining the ranking-based linear reward as $R_k = a \cdot r_k + b$, the GRPO group normalization strictly simplifies the advantage function to:
> > >
> > > $$\tilde{A}_k = \frac{(a \cdot r_k + b) - (a \cdot \mu_r + b)}{|a| \cdot \sigma_r} = \text{sgn}(a) \cdot \frac{r_k - \mu_r}{\sigma_r}$$
> > >
> > > This proves the optimization gradient relies solely on relative rankings, completely independent of absolute score shifts or scaling. It forces the model to focus on true global rankings, thereby mechanistically eradicating representation collapse caused by score inflation.
> > >
> > > **2. Variance Lower Bound:** Traditional continuous scoring often causes reward variance to approach zero ($\sigma \to 0$) during homogenous exploration, risking gradient explosion. In contrast, our discrete ranking variable ($r_k \in \mathbb{Z}^+$) guarantees a strictly positive minimum variance ($\sigma_{\min} > 0$), even when intra-group rewrites are highly similar. This theoretical "variance floor" fundamentally eliminates the pathological risk of vanishing denominators in RL training.
> > >
> > > Ultimately, these theoretical guarantees go beyond ensuring numerical stability. They demonstrate that our method robustly bridges the modality gap between user queries and pre-trained visual spaces without altering the frozen CLIP-like retriever. This provides both a practical solution to real-world query distribution mismatches and also serves as a theoretical inspiration for the in-depth application of reinforcement learning in cross-modal alignment in the future.
> > >
> > > Finally, we promise to comprehensively revise all details discussed in this Rebuttal in the camera-ready version of the paper.

---

### Official Review · Reviewer_fMh1 · 2026-03-12

**Soundness:** 3
**Presentation:** 3
**Significance:** 3
**Originality:** 3
**Overall Recommendation:** 5
**Confidence:** 3

**Summary:**

The problem this paper targets is the degradation of CLIP-based retrieval performance under several types of query-side distribution shift in multilingual, long-form, multimodal settings. The angle through which they tackle this is to propose a training-free method that uses a LLM to rewrite the query, using a retrieval-derived ranking feedback and a GRPO-style objective.

**Compliance With Llm Reviewing Policy:**

Affirmed.

**Final Justification:**

The rebuttal has helped address my concerns, so I have raised the overall score.

**Key Questions For Authors:**

Q1. Comparison to stronger training-free baselines

The paper positions GRAPE within a broader space of training-free retrieval enhancement methods, but the main comparisons are against frozen CLIP and naive LLM rewriting. Could the authors compare GRAPE against stronger existing training-free baselines (e.g., plug-in retrieval enhancement methods cited in related work)?

Q2. Understanding when GRAPE helps most

The magnitude of improvements varies across datasets. Do the authors have hypotheses about what task characteristics drive larger vs smaller gains, and when GRAPE should be expected to help most in practice?

Q3. Long-form benchmark characterization

The paper uses the Wikipedia dataset to represent long-form query settings. Could the authors provide additional characterization of the query lengths or distribution in this dataset, and comment on how query length relates to the observed gains?

**Limitations:**

yes

**Strengths And Weaknesses:**

**Strengths:**

First, their propose method (GRAPE) shows consistent improvements over basic CLIP (no rewriting) and CLIP + vanilla LLM query rewriting across datasets, showing how the proposed method provides value beyond simple prompting. On top of providing improvements, the method is also easy to adopt as it does not require retraining the retrieve or re-embedding the corpus. This deployment-friendliness allows it to be useful in practice. Finally, they bring up a very interesting insight about the flaws of similarity-based supervision. Namely, it results in an optimization shortcut that generates high-frequency, non-discriminative tokens.

**Weaknesses:**

One of the benefits they claim for their method is that it is training-free. However, there is limited comparison against other training-free retrieval enhancement approaches, like PlugIR mentioned in the related works. Some comparison here can shed light on whether the improvements are specific to GRAPE.

Another aspect that could improve the practical takeaways is some analysis of when and why improvement vary across settings. Although GRAPE consistently improves over the reported baselines, the magnitude of gains differs substantially across datasets. Some potential insight on what characteristics drive larger vs smaller improvements could be enlightening for a practitioner looking to adopt this method.

Lastly, they claim that long-form query is one of the core issues they target, but there is not that much analysis on relating performance to query length. This could help clarify if it helps equally/more/less for shorter length queries for example.

---

> ### Author Rebuttal · Authors · 2026-03-30
>
> **Reviewer fMh1**
> We thank the reviewer for the constructive feedback.
>
> **Q1. Comparison to stronger training-free baselines**
> | Dataset | Claude-Sonnet-4.6 | GPT-5.4 | PlugIR | **GRAPE (Ours)** |
> |:---|:---|:---|:---|:---|
> | Flickr-CN (1k) | 56.2 / 91.3 | 56.2 / 89.0 | 51.4 / 84.4 | **58.7 / 90.1** |
> | Wikipedia | 36.9 / 73.9 | 38.4 / 78.4 | 33.2 / 73.0 | **43.7 / 82.8** |
>
> We add comparisons with a stronger training-free baseline, PlugIR, across two datasets. To ensure a controlled and fair comparison in the training-free setting, we isolate PlugIR’s query rewriting component without interactive feedback. GRAPE consistently outperforms PlugIR.
>
> We attribute this improvement to a key difference: PlugIR remains prompt-driven and lacks explicit supervision from the retriever. As discussed in our paper, prompt-only rewriting is prone to producing semantically plausible but retrieval-suboptimal queries due to the absence of ranking-aware feedback. In contrast, GRAPE directly optimizes rewriting with ranking-based supervision, leading to better alignment with the retriever’s latent space.
>
> We further compare against strong LLM-based rewriting baselines (e.g., GPT5.4 and Claude Sonnet-4.6). Despite using a smaller rewriter (Qwen2.5-3B), GRAPE consistently outperforms these approaches (e.g., 43.7 vs. 38.4 R@1 on Wikipedia), suggesting that the gains stem from retriever-aligned optimization rather than model scale. We will include these comparisons in the revised version.
>
>
> **Q2. Understanding when GRAPE helps most**
>
> We thank the reviewer for this insightful question.
> Empirically, we observe that GRAPE yields larger gains when there is a greater distribution gap between the query and the retriever’s training distribution, which aligns with our method design. This gap typically arises in three scenarios:
>
> 1.  Length shift (e.g., long-form Wikipedia queries), where GRAPE performs semantic distillation and removes redundancy;
> 2. Semantic or compositional complexity (e.g., CIRR), where GRAPE helps recover retrieval-relevant attributes;
> 3. Cross-lingual mismatch (e.g., Flickr30k-CN), where GRAPE enriches implicit semantics beyond direct translation.
>
> Consistent with Table 1 in the paper, these settings show relatively larger improvements over CLIP+LLM baselines.
>
> In contrast, gains are smaller when the improvement is bounded by model capacity. For example, when the LLM lacks sufficient knowledge for low-resource languages (as also discussed in Sec. 5), or when the frozen retriever itself has limited representational coverage, query rewriting alone cannot fully bridge the gap.
>
> Practical takeaway: GRAPE is most beneficial in settings with substantial query–retriever mismatch (e.g., long-form, multilingual, or compositional queries), and less impactful when bottlenecks lie in the retriever or LLM capacity rather than query formulation.
>
> We will clarify these observations and add a “Practical Guidelines” paragraph in the revision.
>
> **Q3. Long-form benchmark characterization**
>
> We have added statistics on query length for the Wikipedia dataset, summarized in the table below, in which LLM (Explicit Shorten) means we explicitly force LLM to generate shorter rewriting results:
>
> | Method | Average Length (Words) | Standard Deviation | Recall@1 / Recall@10 (%) |
> | :--- | :---: | :---: | :---: |
> | Raw Query | 533 | 237 | 33.5 / 71.9 |
> | LLM Rewrite | 91 | 45 | 34.3 / 69.3 |
> | LLM (Explicit Shorten) | 35.7 | 13 | 34.8 / 73.7 |
> | **GRAPE (Ours)** | **45** | **14** | **43.7 / 82.8** |
>
> As shown above, GRAPE produces shorter and more stable queries, partially aligning with the retriever’s preferred input distribution.
>
> However, length alone does not explain the gains: enforcing even shorter outputs (35.7 words) results in significantly worse performance (34.8 / 73.7) compared to GRAPE (43.7 / 82.8). This indicates that the improvement comes not merely from shortening, but from generating concise yet retrieval-oriented rewrites that preserve key entities and structure. We will include this analysis in the revised version.

---

> > ### Author Rebuttal · Reviewer_fMh1 · 2026-04-04
> >
> > The rebuttal addresses my main concerns by adding comparisons with stronger training-free baselines (e.g., PlugIR) and providing a clearer characterization of when GRAPE is most effective (e.g., under query–retriever mismatch such as long-form or multilingual queries). The additional analysis on query length further clarifies that the gains are not due to simple shortening but to retrieval-aligned rewriting. I will be adjusting the scores accordingly in the final justification form.

---

> > > ### Author Response · Authors · 2026-04-07
> > >
> > > Thank you so much for your careful review of our rebuttal and your constructive feedback throughout the review process. We greatly appreciate your recognition of our responses and revisions in the rebuttal, and we are truly grateful for your time, effort, and your decision to adjust the scores accordingly. We will carefully refine the final manuscript following all your valuable suggestions.

---

### Decision · Program_Chairs · 2026-04-30

**Decision:**

Accept (regular)

**Comment:**

This paper addresses a practical and important problem, namely,  improving frozen CLIP-based retrieval under multilingual, long-form, and multimodal query shifts without retraining the retriever or re-embedding the corpus. Reviewers generally agree that the paper is technically solid, clearly written, and supported by strong empirical results. The rebuttal successfully addressed most of the empirical and practical concerns, including comparisons with stronger training-free baselines and retrievers, query-length analysis, robustness across backbones, and discussion of stability and failure modes. The remaining disagreement mainly concerns novelty rather than soundness. While some reviewers noted that ranking-based supervision for query rewriting is already known in the retrieval literature, the authors’ follow-up response clarified that the contribution goes beyond a simple engineering adaptation. Overall, although opinions may still differ on whether the level of novelty is sufficient, the authors have responded substantively to the main criticism. Given the practical importance of the problem, the solid experimental support, and the strengthened clarification of the contribution, my recommendation is Weak Accept.